# Morusin Protected Ruminal Epithelial Cells against Lipopolysaccharide-Induced Inflammation through Inhibiting EGFR-AKT/NF-κB Signaling and Improving Barrier Functions

**DOI:** 10.3390/ijms232214428

**Published:** 2022-11-20

**Authors:** Chunlei Yang, Xiangfei Deng, Linjun Wu, Tianrui Jiang, Zhengwei Fu, Jinjun Li

**Affiliations:** 1College of Biotechnology and Bioengineering, Zhejiang University of Technology, Hangzhou 310014, China; 2Institute of Food Sciences, Zhejiang Academy of Agricultural Sciences, Hangzhou 310021, China

**Keywords:** Morusin, ruminal epithelial cells, immune regulation, EGFR, AKT, NF-κB p65

## Abstract

Using phytogenic extracts for preventing or treating rumen epithelial inflammatory injury is a potential alternative to antibiotic use due to their residue-free characteristics. In this study, the efficacy of *Morus* root bark extract Morusin on ruminal epithelial cells (RECs) against pathogenic stimulus was investigated for the first time. The 3-(4,5-dimethylthiazol-2-yl)-2,5-diphenyltetrazolium bromide (MTT) and quantitative real-time polymerase chain reaction (qPCR) results showed that the Morusin did not affect the cell viability of RECs and exerted anti-inflammatory effects in a concentration-dependent manner. Transcriptome analysis further revealed that the Morusin significantly downregulated the inflammatory-response-related cell signaling, while it upregulated the cell-proliferation-inhibition- and barrier-function-related processes in RECs upon lipopolysaccharide (LPS) stimulation. The epidermal growth factor receptor (EGFR) blocking and immunoblotting analysis further confirmed that the Morusin suppressed LPS-induced inflammation in RECs by downregulating the phosphorylation of protein kinase B (AKT) and nuclear factor-kappaB (NF-κB) p65 protein via inhibiting the EGFR signaling. These findings demonstrate the protective roles of Morusin in LPS-induced inflammation in RECs.

## 1. Introduction

As the major digestive site of ruminants, ruminal epithelium not only plays a significant role in nutrient absorption and transportation, but also forms a tight barrier against endotoxins that may derive from ruminal microorganisms and feed [1]. The free lipopolysaccharide (LPS) released from the shedding gram-negative bacteria or lysis of dead microbial cells within the rumen usually represents a significant challenge for ruminal epithelium, particularly when coinciding with rumen dysfunction, such as ruminal acidosis [2]. The increased LPS concentration in the rumen has been reported to substantially contribute to the induction of inflammatory responses in ruminal epithelium, which may endanger its integrity and barrier function and further allow the translocation of LPS into blood and cause systemic inflammation [3].

In consideration of the increasing public awareness about food safety and the environmental impact related to antibiotic use and animal welfare, developing phytogenic alternatives for the prevention and treatment of inflammatory diseases has attracted much attention due to their “natural, safe, and residue-free” characteristics [4,5]. The root bark of *Morus*, which is rich in various bioactive compounds such as flavonoids, triterpenoids, alkaloids, and steroids, has long been used in herbal medicine as liver- and kidney-protective, anti-phlogistic, antioxidant, immunomodulatory, anti-tumor, anti-bacterial, and anti-hypotensive agents [6,7]. Among the various classes of prenylated flavonoids that have been isolated from *Morus* root bark, Morusin is highlighted for its versatile influences on human pathology and physiology due to its unique physiochemical properties [8]. Past studies indicated that Morusin can suppress inflammatory signaling in stimulated keratinocytes via inhibiting the signal transducer and activator of transcription 1 (STAT1) and nuclear factor-kappaB (NF-κB) p65 phosphorylation [9], reduce cell proliferation and promote cell apoptosis of osteosarcoma cells by inhibiting the PI3K-AKT signaling pathway [10], and alleviate chemical-induced rat colitis by reducing the production of proinflammatory and fibrosis markers (IL-1β, TGF-β, MMP-2 and 9) [11]. Using the docking analysis, Park et al. [12] proposed that Morusin may have the potential to bind to the catalytic domain of epidermal growth factor (EGF) receptor (EGFR) for its inactivation, which perhaps mediates its inhibitory effects on inflammation and associated abnormal cell growth. However, the specific mechanisms explaining the anti-inflammatory effects of Morusin have not been fully elucidated.

Using flavonoid-rich plant extracts to improve the immunity and health of ruminants has been previously tested. For example, Zhan et al. [13] found that the antioxidant status of dairy cows was significantly improved by the supplementation of alfalfa flavonoids. Liu et al. [14] found that the addition of moringa oleifera leaf flavonoids effectively inhibited hydrogen-peroxide-induced oxidative stress in bovine mammary epithelial cells and reduced the accumulation of reactive oxygen species. However, the potential regulatory effects of *Morus*-derived flavonoid Morusin on protecting ruminal epithelium against pathogenic stimulus are rarely reported. The expression profile of eukaryotic genes and proteins plays an essential role in multiple biological processes and thus regulates the emergence of specific phenotypes, and in-depth study of genes and proteins at the cellular level has enabled the possibility of elucidating the mechanisms by which exogenous factors influence epigenetic traits [15]. Therefore, to better understand the possible immunoregulatory effects of Morusin on ruminal epithelial cells (RECs) and to elucidate the related molecular mechanisms, the changed expression and function of genes and proteins in RECs pretreated with Morusin under inflammatory conditions were comprehensively investigated in this study.

## 2. Results

### 2.1. Morusin Alleviated LPS-Induced RECs’ Inflammation in a Concentration-Dependent Manner

Characterization of the RECs used in present study certified their validity after 33 passages (Appendix A). Firstly, we examined whether Morusin is toxic to RECs using an MTT assay. The results showed that neither treatment with 25 or 50 µg/mL of Morusin alone nor Morusin combined with LPS affected the cell viability of RECs (Figure 1A). Then, qPCR was used to determine the better dose of Morusin for alleviating LPS-induced inflammation in RECs. The results showed that although both the 25 and 50 µg/mL Morusin significantly inhibited the gene expression of *NF-κB* (*p* < 0.01 and *p* < 0.001) and *IL-1β* (*p* < 0.05 and *p* < 0.01) in RECs upon LPS stimulation (Figure 1B,C), the regulatory effects of Morusin on the gene expression of proinflammatory cytokine *TNF-α*, TNF receptor *CD40*, and *IL-6*, as well as chemokine *CCL20* in RECs upon LPS stimulation were concentration-dependent. The pretreatment of 25 µg/mL Morusin for 12 h did not significantly affect the gene expression of *TNF-α*, *CD40*, *IL-6*, or *CCL20* in RECs upon LPS stimulation (Figure 1D–G). However, the pretreatment of 50 µg/mL of Morusin for 12 h significantly downregulated the gene expression of *TNF-α* (*p* < 0.001), *CD40* (*p* < 0.05), *IL-6* (*p* < 0.001), and *CCL20* (*p* < 0.05) in RECs upon LPS stimulation (Figure 1D–G). Therefore, 50 µg/mL of Morusin for 12 h exhibited better protective effects against LPS-induced inflammation in RECs and was further used for transcriptomics and protein expression analysis to determine its regulatory mechanism.

### 2.2. Overview of the Transcriptome in RECs among Different Treatments

An average of 46.13 million clean reads per sample were obtained from transcriptome sequencing after raw data cleaning. Through calculating FPKM (fragments per kilobase of transcript per million fragments mapped) of all the genes that mapped to *Ovis aries* in each sample, a similar gene expression distribution was observed. After comparing the differentially expressed genes (DEGs) (*p*_adj_ < 0.05 and fold change > 2) among all the treatments, there were 1507 genes significantly upregulated, and 1232 genes were significantly downregulated in the LPS group compared with the expression levels in the NC group. Simultaneously, 1169 genes were significantly upregulated, and 1304 genes were significantly downregulated in the Morusin+LPS group compared with the LPS group. Specifically, 1107 overlapping genes were significantly upregulated with the LPS stimulation alone (LPS vs. NC up) and downregulated with the pretreatment of Morusin upon LPS stimulation (Morusin+LPS vs. LPS down), while 978 overlapping genes were significantly downregulated with the LPS stimulation alone (LPS vs. NC down) and upregulated with the pretreatment of Morusin upon LPS stimulation (Morusin+LPS vs. LPS up) (Figure 2A). Hierarchical cluster analysis of these overlapping DEGs further showed that the gene expression profile in the NC and Morusin+LPS groups were clearly separated from the LPS group (Figure 2B).

### 2.3. Morusin Altered Cell Signaling to Regulate LPS-Induced Inflammation in RECs

To test the potential regulatory mechanisms of Morusin on alleviating LPS-induced inflammation in RECs, Gene Ontology (GO) and Kyoto Encyclopedia of Genes and Genomes (KEGG) enrichment analysis were conducted for the overlapping DEGs to define their biological functions. The results showed that a total of 76 GO terms were significantly enriched from the overlapping genes that were downregulated in the Morusin+LPS group compared with the LPS group and upregulated in the LPS group compared with NC group, and they were primarily enriched in the biological processes, including cytokine receptor binding, growth factor receptor binding, interleukin-1 receptor binding, cytokine activity, and response to endoplasmic reticulum stress (*p* < 0.05) (Figure 3A). KEGG analysis further showed that 20 pathways were significantly enriched from these overlapping genes, including the NOD-like receptor signaling pathway, TNF signaling pathway, IL-17 signaling pathway, and cytokine–cytokine receptor interaction (*p* < 0.05) (Figure 4A).

For the overlapping genes that were upregulated in the Morusin+LPS group compared with LPS group and downregulated in the LPS group compared with NC group, 688 GO terms were significantly enriched, which primarily involved biological processes such as actin binding, integrin binding, cell–substrate adherens junction, anchoring junction, and extracellular structure organization (*p* < 0.05) (Figure 3B). According to KEGG analysis of these overlapping genes, 42 pathways were significantly enriched, which primarily involved gap junction, adherens junction, tight junction, ECM–receptor interaction, regulation of actin cytoskeleton, and Hippo signaling pathway (*p* < 0.05) (Figure 4B).

### 2.4. Morusin Altered the Expression of Hub Genes Involved in Inflammation, Cell Proliferation, and Cell Junction Processes in RECs

Protein–protein interaction (PPI) network analysis was conducted to test the potential interactions among the overlapping genes. In total, 149 nodes and 879 edges were established in the PPI network for the overlapping genes that were enriched when comparing Morusin+LPS vs. LPS down and LPS vs. NC up, as well as Morusin+LPS vs. LPS up and LPS vs. NC down. The genes shown in Figure 5A were identified as hub genes after degree centrality analysis (degree > 10). Among them, the inflammation-related genes including *IL-6*, *CSF2*, *IL-1A*, *CCL20*, *CXCL6*, and *CXCL8* and the cell-proliferation-related genes including *RIPK2* and *ATF4* were significantly upregulated in the LPS group compared with those in the NC group; however, the pretreatment with Morusin significantly downregulated these genes in RECs upon LPS stimulation (*p* < 0.05) (Figure 5B). On the other hand, the cell-proliferation-inhibition-related genes including *ACTB* and *FZD2* and cell-junction-related genes including *ACTN1* and *VCL* were significantly downregulated in the LPS group compared with those in the NC group, whereas the pretreatment with Morusin significantly upregulated these genes in RECs upon LPS stimulation (*p* < 0.05) (Figure 5B).

### 2.5. Morusin Inhibited the Activation of AKT and NF-κB Signaling in RECs upon LPS Stimulation

AKT and NF-κB signaling are known to be key regulators in LPS-induced inflammatory responses and abnormal cell proliferation. Therefore, we examined whether Morusin regulated LPS-induced phosphorylation (p) of AKT, IκBα, and NF-κB p65 in RECs using immunoblotting. The results showed that the LPS stimulation significantly upregulated the protein expression levels of p-AKT/AKT (*p* < 0.05), p-IκBα/IκBα (*p* < 0.01), and p-p65/p65 (*p* < 0.05), while it downregulated the protein expression of IκBα/β-actin (*p* < 0.05) in RECs compared with the negative control. However, the pretreatment of Morusin significantly downregulated the protein expression levels of p-AKT/AKT (*p* < 0.05), p-IκBα/IκBα (*p* < 0.01), and p-p65/p65 (*p* < 0.05) in RECs upon LPS stimulation (Figure 6).

### 2.6. Morusin Suppressed LPS-Induced AKT and NF-κB Activation by Inhibiting EGFR-Mediated Signaling

Cell surface receptor EGFR is suggested to be a significant upstream regulator of AKT and NF-κB signaling, and Morusin has the potential to bind to the catalytic domain of EGFR for its inactivation [12]. Thus, we tested whether the inhibitory effects of Morusin on the AKT and NF-κB signaling in RECs upon LPS stimulation were mediated by its inhibition of EGFR using AZD-9291. The immunoblotting results showed that the LPS stimulation significantly upregulated the protein expression level of p-EGFR/EGFR in RECs compared with negative control (*p* < 0.01); however, the pretreatment of Morusin significantly downregulated the protein expression of p-EGFR/EGFR during LPS stimulation (*p* < 0.05) (Figure 7). Consistent with our above findings, the pretreatment of Morusin also significantly suppressed the protein expression of p-AKT/AKT, p-IκBα/IκBα, and p-p65/p65 in RECs upon LPS stimulation (*p* < 0.05) (Figure 7). In addition, the combination of AZD-9291, Morusin, and LPS further significantly downregulated the protein expression of p-EGFR/EGFR (*p* < 0.01), as well as p-AKT/AKT (*p* < 0.05), p-IκBα/IκBα (*p* < 0.05), and p-p65/p65 (*p* < 0.01), while it upregulated the IκBα/β-actin (*p* < 0.05) in RECs compared with the treatment of Morusin and LPS (Figure 7). Simultaneously, the combination of AZD-9291, Morusin, and LPS also further significantly suppressed the protein expression of p-EGFR/EGFR (*p* < 0.05), p-AKT/AKT (*p* < 0.05), p-IκBα/IκBα (*p* < 0.05), and p-p65/p65 (*p* < 0.05), while it upregulated the IκBα/β-actin (*p* < 0.05) in RECs compared with the treatment of AZD-9291 and LPS (Figure 7).

## 3. Discussion

Ruminal epithelium is the first line of defense in the rumen digestive system and initiates immune responses to pathogens and noxious irritants. It must always face the challenges of harmful antigens within the rumen, such as the free LPS derived from shedding gram-negative bacteria and the lysis of dead microbial cells, particularly when coinciding with pathological conditions such as ruminal acidosis [1,2]. The abnormally activated rumen epithelial cells produce a variety of inflammatory mediators and cytokines, which is crucial in mediating further ruminal epithelial barrier disruption [3]. In consideration of the adverse effects of antibiotics, using “natural, safe, and residue-free” substances for protecting rumen immune homeostasis has attracted much attention [5]. Morusin is a kind of isoprene flavonoid isolated from the Chinese herbal medicine *Morus* that has shown beneficial effects on immunoregulation [8]. Our present study demonstrated that 50 µg/mL Morusin, rather than 25 µg/mL, could effectively alleviate LPS-induced inflammation in RECs via suppressing the production levels of proinflammatory cytokines, including TNF-α and IL-6, TNF receptor superfamily member CD40, and chemokine CCL20, which are significantly involved in the induction of cascade amplification inflammatory responses [16], highlighting the excellent potential of Morusin as a preventive drug in the healthy farming of ruminants; however, its anti-inflammatory effects are concentration-dependent.

RNA-Seq transcriptome was used in the present study to investigate the potential anti-inflammatory mechanisms of Morusin and found the pretreatment of Morusin primarily downregulated the inflammatory pathways involved in the NOD-like receptor (NLR), TNF and IL-17 signaling, as well as the cytokine–cytokine receptor interaction in RECs upon LPS stimulation. It is known that the specific host pathogen recognition receptors (PRRs) play significant roles in recognizing conserved pathogen-associated molecular patterns like LPS [17]. The innate immune receptor NLR is a kind of PRR that can translate danger recognition into a wide range of inflammatory responses, including the production of proinflammatory cytokines and chemokines, with the attendant recruitment of neutrophils or other immune cells, as well as cell death [18]. Growing evidence has supported that the NLR-mediated signaling is a critical factor involved in LPS-stimulated inflammatory responses. For example, Wu et al. [19] found that LPS simulation could promote inflammatory NF-κB signaling in an alveolar macrophage by activating the NLR family pyrin domain containing 3 (NLRP3) inflammasome, with large amounts of cytokines produced, such as IL-1β and IL-18. The significant role of TNF-α and IL-17 in amplifying and perpetuating inflammatory damage from NLRP3 activation has been certified in the study of Wree et al. [20] via breeding NLRP3 knock-in mice onto the IL-17α or TNF knockout background, and they found that the recruitment of neutrophils or circulating IL-1β levels in NLRP3 mutants were significantly inhibited by the ablation of IL-17α or TNF. In addition, the inflammatory cytokines and chemokines produced by the activated inflammatory cell signaling are proposed to be key components in modulating the initiation and progression of barrier disruption. For example, increased TNF-α production has been reported to closely associate with the LPS-induced impairment of colon barrier function in mice [21]. In the present study, corresponding to the significantly downregulated inflammatory cell signaling, the epithelial-barrier-related signaling, including gap junction, tight junction, adherens junction, ECM–receptor interaction, regulation of actin cytoskeleton, actin and integrin binding, and extracellular structure organization [22], as well as the expression of cell-junction-related genes including *ACTN1* [23] and *VCL* [24], were all significantly upregulated in RECs upon LPS stimulation with the Morusin pretreatment. Therefore, our findings suggest that Morusin may protect RECs against LPS stimulation via inhibiting the NOD-like-receptor-mediated activation of TNF and IL-17 signaling pathways and enhancing barrier-function-related cell signaling.

Emerging evidence suggests that NF-κB and AKT signaling are key associated transcription factors involved in LPS-induced inflammatory responses and abnormal cell proliferation, even considered as prospective targets for anti-inflammation [25]. NF-κB can regulate the expression of various genes related to proliferation, apoptosis, and inflammation [26]. Under inflammatory conditions such as LPS stimulation, the IκBα is activated by phosphorylation and then ubiquitinated and degraded by 26S proteasome, which further results in the release of NF-κB, including p65 from the cytoplasmic NF-κB–IκBα complex and leads to its subsequent phosphorylation and nuclear translocation. Translocated NF-κB p65 finally activates the expression of target genes involved in proinflammatory responses such as *IL-1β*, *IL-6*, and *TNF-α* and cell proliferation such as *cyclin D1* and *c-Myc* [27,28]. Signal transduction by AKT, another main kinase, is also predominantly responsible for proinflammatory cytokine releases and cell growth regulation [25]. For example, Du et al. [29] found that the activation of AKT signaling is required for the TNF-α-induced proinflammatory responses and abnormal cell proliferation of synovial fibroblasts, while blocking the AKT pathway could effectively prevent the production of inflammatory factors and inhibit cell proliferation when coinciding with TNF-α stimulation. In addition, Tang et al. [30] found that the knockout of AKT significantly reduced the production of proinflammatory cytokines in human liver cancer cell lines via inhibiting NF-κB activation. Therefore, the modulation of AKT and NF-κB signaling may play a crucial role in the control of inflammatory responses. Interestingly, our study verified that pretreatment with Morusin significantly suppressed the phosphorylation of IκBα, NF-κB p65, and AKT in RECs upon LPS stimulation. Correspondingly, the gene expressions of proinflammatory cytokines *IL-6*, *CSF2*, and *IL-1A* [31], chemokines *CCL20*, *CXCL6*, and *CXCL8* [32], and cell proliferation factor *RIPK2* [33] and *ATF4* [34] were significantly downregulated, whereas the expression of genes related to cell proliferation inhibition, including *ACTB* [35] and *FZD2* [36], as well as cell-proliferation-inhibition-related Hippo signaling [37] were significantly upregulated in LPS-induced RECs with Morusin pretreatment, which underlines the fact that Morusin can protect RECs against LPS-induced inflammation and abnormal cell proliferation via inhibiting the AKT and NF-κB signaling pathways.

Given that the activation of the AKT or NF-κB signaling pathway is the downstream event of EGFR signaling and that Morusin has the potential to directly bind to the catalytic domain of EGFR for its inactivation [8,38], we used the EGFR inhibitor to confirm whether the anti-inflammatory effects of Morusin on LPS-induced inflammation in RECs was mediated by the downregulation of AKT or NF-κB signaling via EGFR inhibition. EGFR is a kind of tyrosine kinase receptor that is necessary for the proliferation and activation of inflammatory cells [39]. Epithelial EGFR activation has been found to significantly contribute to neutrophilic inflammation, mucous metaplasia, and secretion of inflammatory mediators [40,41], while the inhibition of EGFR can effectively attenuate irritant-induced epithelial resistance and proinflammatory cytokine secretion as well as improve the restoration of epithelial junctions [42]. Previous studies have also shown that the treatment of EGFR inhibitors can suppress the abnormal cell proliferation of epithelial cancer cell lines by blocking the AKT pathway [38] as well as reduce the expressions of proinflammatory cytokines such as IL-6 via inhibiting the activation of the NF-κB pathway [43]. In the present study, the combination of EGFR inhibitor AZD-9291, Morusin, and LPS further decreased the phosphorylation of EGFR, AKT, IκBα, and NF-κB p65 in RECs compared with the treatments with Morusin and LPS, as well as AZD-9291 and LPS, respectively, suggesting that the inhibitory effects of AKT and NF-κB signaling in RECs upon LPS stimulation after Morusin pretreatment were mediated by the inhibition of EGFR signaling. In conclusion, our study demonstrated that Morusin could protect RECs against LPS stimulation by suppressing inflammatory-response- and cell-proliferation-related cell signaling, while enhancing barrier-function-related cell signaling. Specifically, Morusin alleviated LPS-induced inflammation in RECs by downregulating the AKT and NF-κB signaling pathways via EGFR signaling inhibition (Figure 8). These findings provide new insights to understand the mechanisms underlying the protective effects of Morusin on RECs against inflammatory stimuli and will offer important therapeutic references for its potential use in animal health breeding.

## 4. Materials and Methods

### 4.1. Cell Culture

The REC cell line used in the present study was primarily isolated from the rumen epithelial tissue of Hu lamb and immortalized by our research group. Generally, the immortalized RECs (33 passages) were cultured in DMEM (Gibco, New York, NY, USA), including 2% FBS, 1% epithelial cell additive, and 1% penicillin/streptomycin at 37 °C in a 5% CO_2_ incubator, as previously described [44]. To check the validity of the RECs used in the present study, light microscopy was used to identify their characteristics, and immunofluorescence staining was performed to identify the presence of cytokeratin 19, ZO-1 and vimentin as described previously [44].

### 4.2. Cell Viability Analysis

When REC monolayers reached 70–80% confluency, the cells were treated without (negative control group, NC) (*n* = 4) or with 0.1 µg/mL LPS (from *Escherichia coli* O111:B4, Sigma-Aldrich, Shanghai, China) for 3 h after pretreating with 25 or 50 µg/mL of Morusin (CAS No. 62596-29-6, Chengdu Alfa Biotechnology Co., Ltd., Chengdu, China) for 12 h or not (LPS group, LPS) (*n* = 4). The cytotoxicity of 25 or 50 µg/mL Morusin to RECs when added alone (L-Morusin group, L-Morusin; or H-Morusin group, H-Morusin) (*n* = 4, respectively) or with LPS (L-Morusin+LPS group, L-Morusin+LPS; or H-Morusin+LPS group, H-Morusin+LPS) (*n* = 4, respectively) were determined using the MTT Bioassay Kit (Sangon Biotech, Shanghai, China).

### 4.3. RNA Extraction and Quantitative Real-Time PCR (qPCR)

Total RNA was extracted from the treated cells (NC, LPS, L-Morusin+LPS and H-Morusin+LPS, respectively) (*n* = 4, respectively) using the RNA Pure Kit (Aidlab Biotechnologies Co., Ltd., Beijing, China) according to the manufacturer’s instruction. cDNA was further synthesized using the PrimeScript RT Reagent Kit (Takara, Dalian, China), and qPCR was then performed with SYBR green in the ABI 7500 (Life Technologies, Singapore) using the previously described procedures [45]. The results were finally normalized to the expression of housekeeping genes tyrosine 3-monooxygenase/tryptophan 5-monooxygenase activation protein zeta (YWHAZ) and glyceraldehyde 3-phosphate dehydrogenase (GAPDH) using the 2^-∆∆Ct^ method [45]. All the primers used were designed using the Basic Local Alignment Search Tool [BLAST; National Center for Biotechnology Information (NCBI), Bethesda, MD, USA] and outlined in the Appendix A.

### 4.4. Transcriptome Sequencing

Transcriptome sequencing was conducted for the RECs that were treated without (NC) (*n* = 4) or with 0.1 µg/mL LPS for 3 h after pretreating with 50 µg/mL of Morusin for 12 h (Morusin+LPS) (*n* = 4) or not (LPS) (*n* = 4). Total RNA was extracted from the treated cells, the purity and concentration of extracted RNA was determined using the NanoPhotometer spectrophotometer (IMPLEN, Westlake Village, CA, USA) and Qubit RNA Assay Kit with a Qubit 2.0 fluorometer (Life Technologies, Carlsbad, CA, USA), respectively. The RNA integrity was simultaneously measured using the RNA Nano 6000 Assay Kit with the Bioanalyzer 2100 system (Agilent Technologies, Santa Clara, CA, USA). A total of 3 µg of RNA per sample was then used for library preparation using the NEBNext Ultra RNA Library Prep Kit for Illumina (NEB, Ipswich, MA, USA) following the manufacturer’s protocol. Paired-end sequencing (150 bp) was further performed using the Illumina HiSeq 4000 instrument, with a minimum depth of 40 million reads per sample finally obtained. The transcriptome sequencing work was supported by the Beijing Novogene Biological Information Technology Co., Ltd. (Beijing, China).

### 4.5. Bioinformatics Analysis

Clean data were obtained from the sequenced raw data by removing the reads containing adapter and poly-N sequences with quality control, which were then mapped to the reference genome (ftp://ftp.ensembl.org/pub/release-95/fasta/ovis_aries/) (accessed on 20 March 2022) and reference gene annotations (ftp://ftp.ensembl.org/pub/release-95/gtf/ovis_aries/) (accessed on 20 March 2022) of *Ovis aries* using HISAT, respectively [46]. FPKM calculation was used to evaluate the gene expression level according to the length of the gene and read count mapped to that gene. The DEGs among the NC, LPS, and Morusin+LPS groups were analyzed using the DESeq2 [47], and Benjamini and Hochberg’s approach was used to adjust the *p*-value (*p*_adj_) to control the false discovery rate. The threshold of *p*_adj_ < 0.05 and fold change > 2 was used to filter the DEGs. GO enrichment and KEGG pathway analysis of the DEGs were further conducted to better annotate their biological functions using the GOseq and KOBAS, respectively [48,49]. The GO terms or KEGG pathways with a *p*-value < 0.05 were considered significantly enriched for the DEGs. To better understand the relationships between the identified genes and proteins, PPI network analysis was performed using the STRING v10.5 database (http://string-db.org/) (accessed on 12 April 2022), and hub genes (degree > 10) were determined using the Cytoscape [50]. The sequences obtained in this study were deposited in the NCBI Sequence Read Archive under accession number PRJNA830995.

### 4.6. Immunoblotting Analysis

Western blot was further conducted for the RECs that were treated without (NC) (*n* = 4) or with 0.1 µg/mL LPS for 3 h after pretreating with 50 µg/mL of Morusin for 12 h (Morusin+LPS) (*n* = 4) or not (LPS) (*n* = 4). To obtain the total protein cell lysates, RECs were homogenized and sonicated in the lysis buffer radio-immunoprecipitation assay (RIPA) (Millipore, Billerica, MA, USA) with protease and phosphatase inhibitors (Roche Diagnostics, Basel, Switzerland) added. After measuring the protein concentration using the BCA Kit (Thermo Scientific, Waltham, MA, USA), a total of 30 μg of protein per lane was loaded onto Bis-Tris SDS-PAGE gels (CWBIO, Taizhou, China), and then immunoblots were performed as previously described [51]. The primary antibodies used were anti-EGFR (phospho Y1068) (1:5000, Abcam, Cambridge, UK, ab40815, RRID: AB_732110), anti-EGFR (1:5000, Abcam, ab52894, RRID: AB_869579), anti-phospho-AKT (Ser473) (1:2000, Cell Signaling Technology, Danvers, MA, USA, #4060, RRID: AB_2315049), anti-AKT (1:1000, Cell Signaling Technology, #9272, RRID: AB_329827), anti-IκB alpha (phospho S36) (1:5000, Abcam, ab133462, RRID: AB_2801653), anti-IκB alpha (1:5000, Abcam, ab32518, RRID: AB_733068), anti-NF-κB p65 (phospho S536) (1:1000, Abcam, ab76302, RRID: AB_1524028), anti-NF-κB p65 (1:1000, Abcam, ab32536, RRID: AB_776751), and anti-β-Actin (1:1000, Cell Signaling Technology, #4967, RRID: AB_330288).

### 4.7. Analysis of EGFR Inhibition on the Phosphorylation of AKT and NF-κB p65

When the REC monolayers reached 70–80% confluency, the cells were pretreated with EGFR inhibitor, the AZD-9291 (APExBIO Technology LLC, Shanghai, China), at a concentration of 500 nM or vehicle (1% DMSO) for 1 h, followed by treating without or with 0.1 µg/mL LPS for 3 h after pretreating with 50 µg/mL of Morusin for 12 h or not (*n* = 4, respectively). Five treatments were contained, and after the treatment, immunoblots targeting the protein expression of p-EGFR/EGFR, p-AKT/AKT, p-IκBα/IκBα, IκBα/β-actin, and p-p65/p65 were performed as mentioned above.

### 4.8. Statistical Analysis

SPSS (v22.0, IBM Corp., Armonk, NY, USA), GraphPad Prism (v9.3.0, GraphPad Software, Inc., San Diego, CA, USA), and Image J (v1.8.0, National Institutes of Health, Bethesda, MD, USA) were used for statistical analyses and visualization. The data for the cell viability, gene, and protein expression were presented as mean ± standard error of the mean (SEM). Student’s *t*-test was used for the comparison of two groups. ANOVA followed by Tukey’s post hoc test was used for the comparison among more than two groups. A statistically significant difference was determined as *p*-value < 0.05.

## Figures and Tables

**Figure 1 ijms-23-14428-f001:**
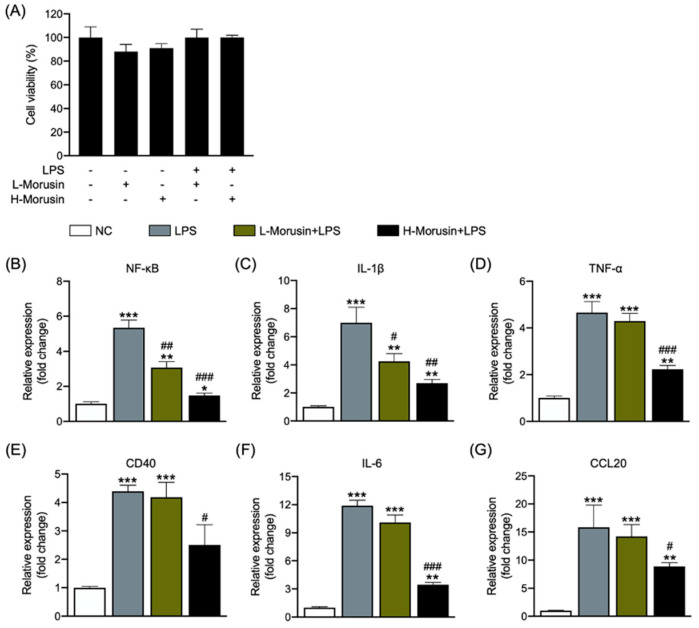
Effects of Morusin on cell viability (**A**) and gene expression of proinflammatory factors *NF-κB* (**B**), *IL-1β* (**C**), *TNF-α* (**D**), *CD40* (**E**), *IL-6* (**F**), and *CCL20* (**G**) in ruminal epithelial cells (RECs) upon lipopolysaccharide stimulation. RECs were treated without (NC, *n* = 4) or with 0.1 µg/mL lipopolysaccharide for 3 h after pretreating with 25 or 50 µg/mL of Morusin for 12 h (L-Morusin+LPS, *n* = 4; H-Morusin+LPS, *n* = 4) or not (LPS, *n* = 4). Data represent mean ± SEM. * *p* < 0.05, ** *p* < 0.01, *** *p* < 0.001 vs. NC; ^#^ *p* < 0.05, ^##^ *p* < 0.01, ^###^ *p* < 0.001 vs. LPS.

**Figure 2 ijms-23-14428-f002:**
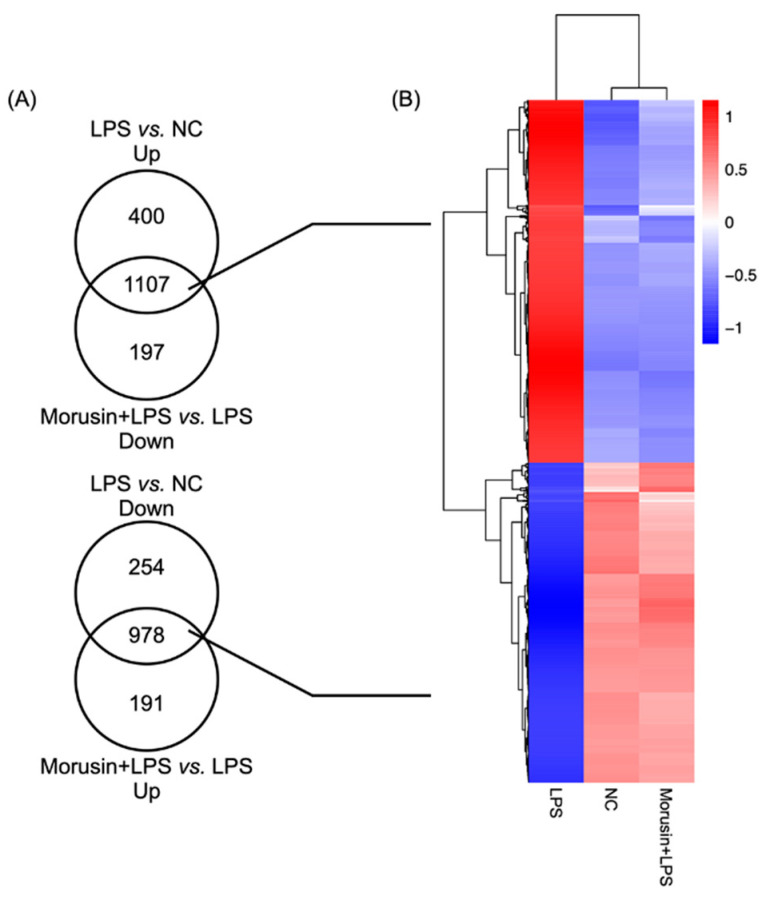
Overview of the differentially expressed genes (DEGs) in ruminal epithelial cells (RECs) among different treatments. (**A**) Venn diagrams of DEGs based on the comparisons of different treatments. Cut-off for the differential expression was determined as *p*_adj_ < 0.05 and fold change > 2. (**B**) Hierarchical clustering of overlapping DEGs among different treatments. The color code indicates the gene expression level, red represents highly expressed genes, and blue represents lowly expressed genes. RECs were treated without (NC, *n* = 4) or with 0.1 µg/mL lipopolysaccharide for 3 h after pretreating with 50 µg/mL of Morusin for 12 h (Morusin+LPS, *n* = 4) or not (LPS, *n* = 4).

**Figure 3 ijms-23-14428-f003:**
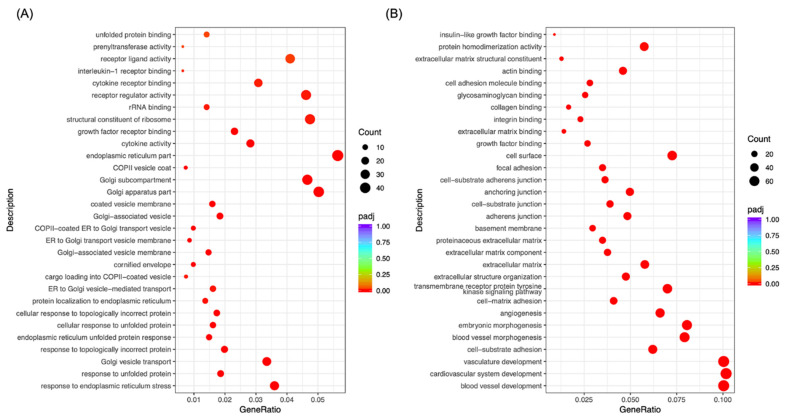
Functional annotation of the differentially expressed genes (DEGs, *p*_adj_ < 0.05 and fold change > 2) in ruminal epithelial cells (RECs) based on Gene Ontology (GO) enrichment analysis. (**A**) GO terms of the overlapping DEGs that were downregulated in the Morusin+LPS group compared with LPS group and upregulated in the LPS group compared with NC group. (**B**) GO terms of the overlapping DEGs that were upregulated in the Morusin+LPS group compared with LPS group and downregulated in the LPS group compared with NC group. Only the top 30 GO terms with a *p*-value < 0.05 were presented. Circles indicate the numbers of enriched genes, colors indicate the *p*_adj_ values. RECs were treated without (NC, *n* = 4) or with 0.1 µg/mL lipopolysaccharide for 3 h after pretreating with 50 µg/mL of Morusin for 12 h (Morusin+LPS, *n* = 4) or not (LPS, *n* = 4).

**Figure 4 ijms-23-14428-f004:**
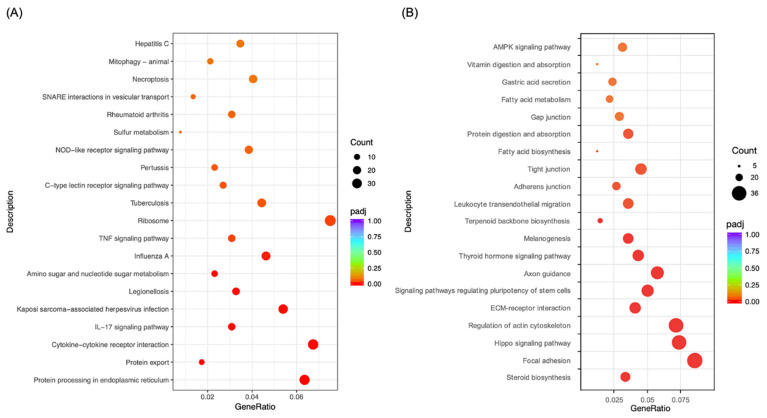
Functional annotation of the differentially expressed genes (DEGs, *p*_adj_ < 0.05 and fold change > 2) in ruminal epithelial cells (RECs) based on Kyoto Encyclopedia of Genes and Genomes (KEGG) pathway analysis. (**A**) KEGG pathway of the overlapping DEGs that were downregulated in the Morusin+LPS group compared with LPS group and upregulated in the LPS group compared with NC group. (**B**) KEGG pathway of the overlapping DEGs that were upregulated in the Morusin+LPS group compared with LPS group and downregulated in the LPS group compared with NC group. Only the top 20 KEGG pathways with a *p*-value < 0.05 were presented. Circles indicate the numbers of enriched genes, and colors indicate the *p*_adj_ values. RECs were treated without (NC, *n* = 4) or with 0.1 µg/mL lipopolysaccharide for 3 h after pretreating with 50 µg/mL of Morusin for 12 h (Morusin+LPS, *n* = 4) or not (LPS, *n* = 4).

**Figure 5 ijms-23-14428-f005:**
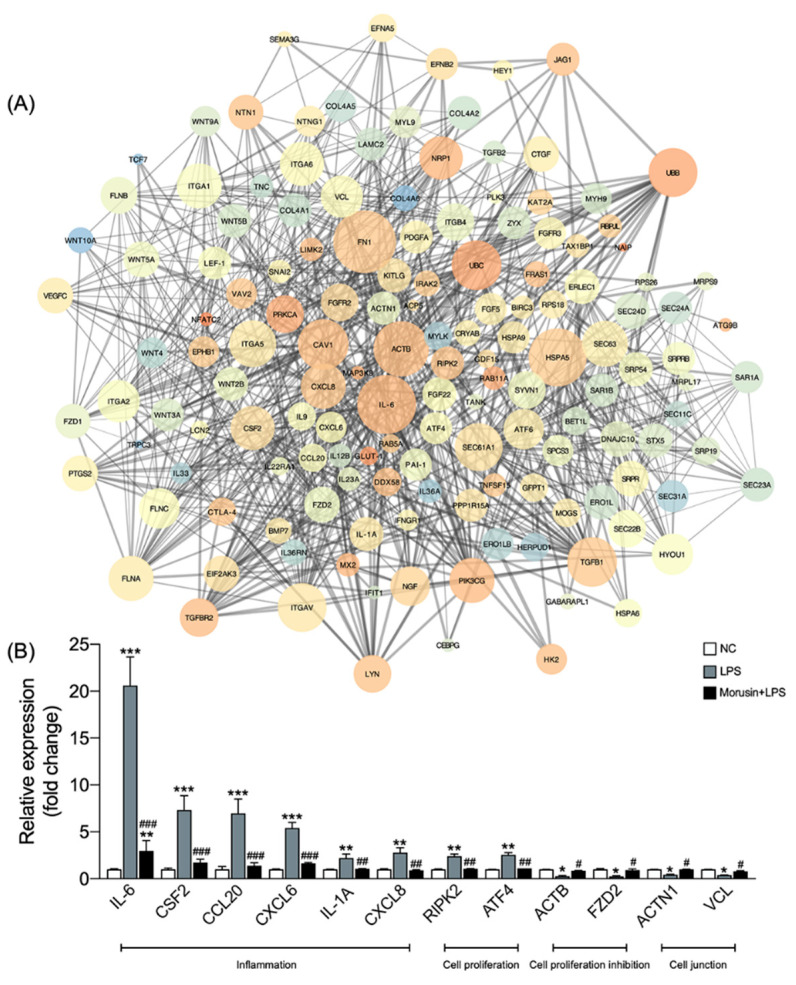
Effects of Morusin on the expression profile of hub genes in ruminal epithelial cells (RECs) with different treatments. (**A**) Protein–protein interaction (PPI) analysis of the overlapping differentially expressed genes (DEGs, *p*_adj_ < 0.05 and fold change > 2) that were downregulated in the Morusin+LPS group compared with LPS group and upregulated in the LPS group compared with NC group, as well as upregulated in the Morusin+LPS group compared with LPS group and downregulated in the LPS group compared with NC group. The node color represents the clustering coefficient (low values are blue, and high values are red), and the node size is proportional to the number of degrees. The edge represents the interaction (the thicker edge indicates the lower betweenness). (**B**) Expression of representative hub genes related to inflammation, cell proliferation, cell proliferation inhibition, and cell junction in ruminal epithelial cells (RECs) among different groups. RECs were treated without (NC, *n* = 4) or with 0.1 µg/mL lipopolysaccharide for 3 h after pretreating with 50 µg/mL of Morusin for 12 h (Morusin+LPS, *n* = 4) or not (LPS, *n* = 4). Data represent mean ± SEM. * *p* < 0.05, ** *p* < 0.01, *** *p* < 0.001 vs. NC; ^#^ *p* < 0.05, ^##^ *p* < 0.01, ^###^ *p* < 0.001 vs. LPS.

**Figure 6 ijms-23-14428-f006:**
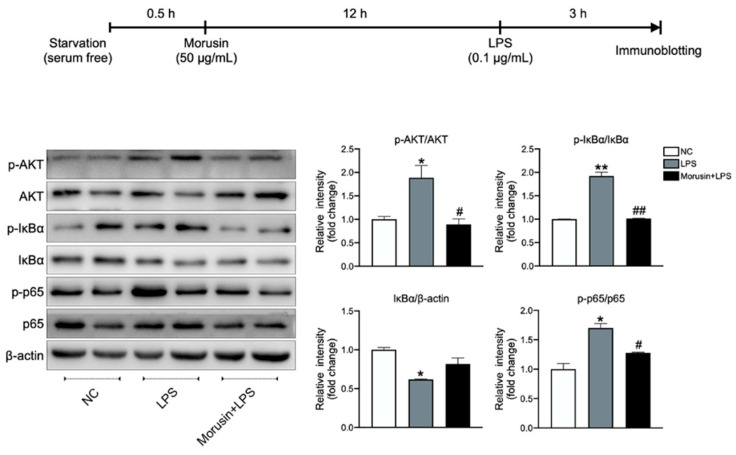
Protein expression level of p-AKT/AKT, p-IκBα/IκBα, IκBα/β-actin, and p-p65/p65 in ruminal epithelial cells (RECs) among different treatments. β-actin was used as reference protein. RECs were treated without (NC, *n* = 4) or with 0.1 µg/mL lipopolysaccharide for 3 h after pretreating with 50 µg/mL of Morusin for 12 h (Morusin+LPS, *n* = 4) or not (LPS, *n* = 4). Data represent mean ± SEM. * *p* < 0.05, ** *p* < 0.01 vs. NC; ^#^
*p* < 0.05, ^##^
*p* < 0.01 vs. LPS.

**Figure 7 ijms-23-14428-f007:**
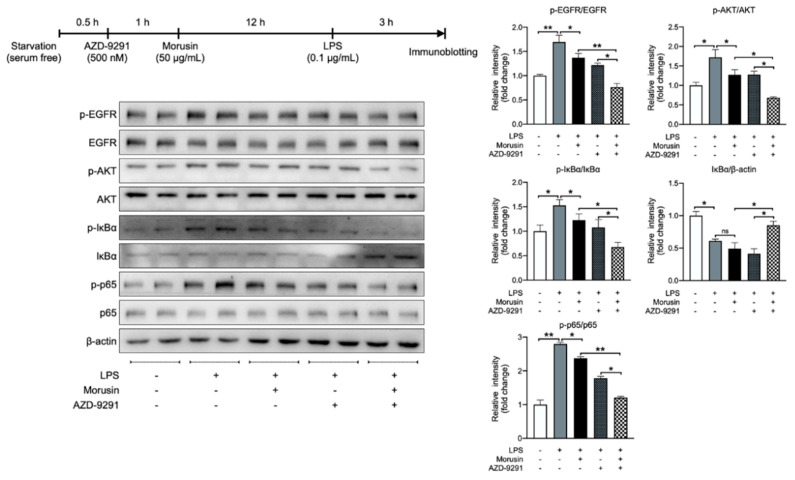
Protein expression level of p-EGFR/EGFR, p-AKT/AKT, p-IκBα/IκBα, IκBα/β-actin, and p-p65/p65 in ruminal epithelial cells (RECs) among different treatments. β-actin was used as reference protein. AZD-9291 is the EGFR inhibitor. RECs were pretreated with AZD-9291 at the concentration of 500 nM or vehicle (1% DMSO) for 1 h, followed by treating without or with 0.1 µg/mL lipopolysaccharide for 3 h after pretreating with 50 µg/mL of Morusin for 12 h or not (*n* = 4, respectively). Data represent mean ± SEM. * *p* < 0.05, ** *p* < 0.01. ns, not significant.

**Figure 8 ijms-23-14428-f008:**
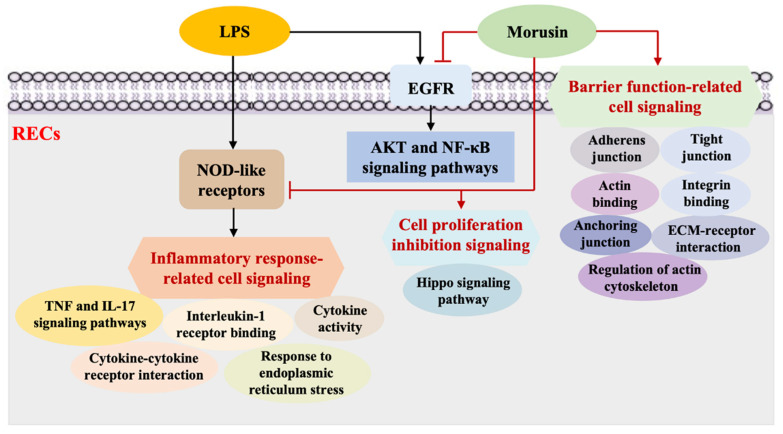
Schematic diagram summarizing the regulatory mechanisms underlying the protective effects of Morusin against lipopolysaccharide (LPS)-induced inflammation in ruminal epithelial cells (RECs). Morusin suppressed the inflammatory-response-related signaling, while it promoted the cell-proliferation-inhibition- and barrier-function-related cell signaling in RECs upon LPS stimulation. Specifically, the pretreatment with Morusin inhibited AKT and NF-κB signaling in RECs upon LPS stimulation by inhibiting EGFR signaling. The arrow represents activation, and the inverted-T represents inhibition.

## Data Availability

The sequences obtained in this study were deposited in the NCBI Sequence Read Archive under accession number PRJNA830995.

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
