# Peer review of "Morusin Protected Ruminal Epithelial Cells against Lipopolysaccharide-Induced Inflammation through Inhibiting EGFR-AKT/NF-κB Signaling and Improving Barrier Functions"

_ijms, 2022, doi:10.3390/ijms232214428_

Round 1

Reviewer 1 Report

Please see comments given in the text of reviewed attached file of manuscript.

Reviewer 2 Report

In this work Authors tested the protective effect Morusin against LPS toxicity in ruminal epithelial cells. They demonstrated that pre-treatment with Morusin is able to counteract inflammation due to LPS exposure inhibiting EGFR-AKT/NF-kB signaling. Additionally, Authors analysed the capability of Morusin to improve barrier functions.

The article is well written and the contents are sound. However, in order to further improve the manuscript, some points should be addressed.

1.     All data sets are analyzed by Student’s t test. However, the Student's t test is used to compare the means between two groups, whereas ANOVA is used to compare the means among three or more groups. Since they analyze the effect of two compounds the right statistical test is one-way ANOVA. Additionally, statistical analyses should be better explained for each figure legend. In particular, in Fig. 5B it is important to describe the meaning of letters (a, b, c) over the bars, i.e. with “a” which groups are compared?

2.     As Authors stated in line 272-273 “the IκBα is activated by phosphorylation and then ubiquitinated and degraded by 26S proteasome”. I will suggest to add also the quantification of IκBα expression level upon treatments in Fig. 6 and 7.

3.     Please add a reference in line 197.

4.     In Fig. 7 the image for p-IκBα should be substituted; the signal is weak and it is not well cropped (it is too in the border of the image).

5.    In 4.5 section, please provide RRID number and the working dilution for each antibody. Additionally, the amount of protein lysate loaded on gels should be indicated.

6. Please check throughout the manuscript for typos, for example line 32 “has” instead of “have” and line 335 “CO2”

Reviewer 3 Report

Even though the paper is easy to read, there are still some questions that need to be clarified.

In line 334, the authors describe the cells as originating from a cell line. Here it is absolutely necessary to name the source of the cell line and, above all, to specify in which passage the experiments were carried out. Since a cell line was used, it is also necessary to show that the characteristics of REC were stable and not lost through the passages.

However, it is confusing that the authors speak of 4 biological replicates in line 342 without further elaboration. Here it is imperative to explain what is meant by the term biological replicates. If 4 different animals are understood as donors, the term “cell line” no longer applies. In addition, in this case too, the extraction and cultivation method must be described clearly and comprehensibly. Cells have also to be characterized.

Against this background, it also remains a mystery how the necessary number of observations was achieved. If one can still assume that Fig. 1A is based on n=4 (line 342), such information is missing for Fig. 1 B/C, 5, 6 and 7. For these data, the information must therefore be provided in the legends. It also remains unclear why the authors used an unpaired T-test and not an analysis of variance with a corresponding post comparison of the individual groups. The statistical procedure suggests that the experiments were not conducted in parallel. If this is the case, however, the presentation in the figures is also obsolete. It has to be clearly explained how many experiments were run in parallel.

In addition, it must be explained how the authors determined a SEM in a relative comparison for the reference variable.

Round 2

Reviewer 3 Report

Unfortunately, the description of the methodological procedure is still very imprecise. The description of checking the validity of the subcultivation is completely missing. In the Figure attached to the author's reply, only a cytokeratin staining of the primary culture is shown. Corresponding images of the subculture must be provided. It must also be shown how other characteristics (for example ZO1, vimentin, fibronectin) of the cell line were checked after the 33rd passage, a pure cytokeratin staining is not sufficient.

An image must be provided from which the 70-80% confluency can be derived (line 369). The figure attached to the author's reply does not support this claim.

The results of the ANOVA should be listed. It should also be clearly stated which approaches were actually carried out in parallel. Only these approaches that were carried out in parallel can be subjected to a statistical comparison - given the authors' approach. A multicomparison test should be applied not a student t test.

The authors must explain what they mean by (independent?) replicates.  

It still remains unclear how a variance/SD of the reference value actually set to 1 was determined. This needs to be clarified.

Round 3

Reviewer 3 Report

The authors have sufficiently addressed the proposals.